**Data Availability Statement:** The datasets used have been uploaded as supplementary information.

**Funding:** This study was conducted with support from Fogarty International Center of the National

# Widespread use of ChatGPT and other Artificial Intelligence tools among medical students in Uganda: A cross-sectional study

Elizabeth Ajalo[1‡], David Mukunya[1‡], Ritah Nantale[1]*, Frank Kayemba[1], Kennedy Pangholi[1], Jonathan Babuya[1], Suzan Langoya Akuu[2], Amelia Margaret Namiiro[3], Yakobo Baddokwaya Nsubuga[4], Joseph Luwaga Mpagi[5], Milton W. Musaba[6], Faith Oguttu[1], Job Kuteesa[7], Aloysius Gonzaga Mubuuke[8], Ian Guyton Munabi[9], Sarah Kiguli[3]

1 Department of Community and Public Health, Busitema University, Mbale, Uganda, 2 Mbarara University of Science and Technology, Mbarara, Uganda, 3 Department of Pediatrics and Child Health, Makerere University, College of Health Sciences, Kampala, Uganda, 4 Faculty of Health Sciences, Gulu University, Gulu, Uganda, 5 Department of Microbiology and Immunology, Busitema University Faculty of Health Sciences, Mbale, Uganda, 6 Department of Obstetrics and Gynaecology, Busitema University Faculty of Health Sciences, Mbale, Uganda, 7 Department of Surgery, Mulago Hospital, College of Surgeons of East Central and Southern Africa, Uganda, 8 Department of Radiology, School of Medicine, College of Health Sciences, Makerere University, Kampala, Uganda, 9 Department of Human Anatomy, School of Biomedical Sciences, Makerere University, College of Health Sciences, Kampala, Uganda

‡ EA and DM are co-first authors on this work.
* ritahclaire24@gmail.com

## Abstract

### Background

Chat Generative Pre-trained Transformer (ChatGPT) is a 175-billion-parameter natural language processing model that uses deep learning algorithms trained on vast amounts of data to generate human-like texts such as essays. Consequently, it has introduced new challenges and threats to medical education. We assessed the use of ChatGPT and other AI tools among medical students in Uganda.

### Methods

We conducted a descriptive cross-sectional study among medical students at four public universities in Uganda from 1st November 2023 to 20th December 2023. Participants were recruited by stratified random sampling. We used a semi-structured questionnaire to collect data on participants' socio-demographics and use of AI tools such as ChatGPT. Our outcome variable was use of AI tools. Data were analyzed descriptively in Stata version 17.0. We conducted a modified Poisson regression to explore the association between use of AI tools and various exposures.

### Results

A total of 564 students participated. Almost all (93%) had heard about AI tools and more than two-thirds (75.7%) had ever used AI tools. Regarding the AI tools used, majority

Institutes of Health, U.S. Department of State's Office of the U.S. Global AIDS Coordinator and Health Diplomacy (S/GAC), and President's Emergency Plan for AIDS Relief (PEPFAR) under Award Number 1R25TW011213. The content is solely the responsibility of the authors and does not necessarily represent the official views of the National Institutes of Health. The funders had no role in study design, data collection and analysis, decision to publish, or preparation of the manuscript.

**Competing interests:** The authors have declared that no competing interests exist.

(72.2%) had ever used ChatGPT, followed by SnapChat AI (14.9%), Bing AI (11.5%), and Bard AI (6.9%). Most students use AI tools to complete assignments (55.5%), preparing for tutorials (39.9%), preparing for exams (34.8%) and research writing (24.8%). Students also reported the use of AI tools for nonacademic purposes including emotional support, recreation, and spiritual growth. Older students were 31% less likely to use AI tools compared to younger ones (Adjusted Prevalence Ratio (aPR):0.69; 95% CI: [0.62, 0.76]). Students at Makerere University were 66% more likely to use AI tools compared to students in Gulu University (aPR:1.66; 95% CI:[1.64, 1.69]).

## Conclusion

The use of ChatGPT and other AI tools was widespread among medical students in Uganda. AI tools were used for both academic and non-academic purposes. Younger students were more likely to use AI tools compared to older students. There is a need to promote AI literacy in institutions to empower older students with essential skills for the digital age. Further, educators should assume students are using AI and adjust their way of teaching and setting exams to suit this new reality. Our research adds further evidence to existing voices calling for regulatory frameworks for AI in medical education.

## Introduction

In November 2022, Open AI launched an artificial intelligence-powered chat box called Chat Generative Pre-trained Transformer (ChatGPT) [1]. Unlike previous versions of internet searching, ChatGPT is a 175-billion-parameter natural language processing model that uses deep learning algorithms trained on vast amounts of data to generate human-like texts such as essays, abstracts, manuscripts, PowerPoint presentations, and book summaries among others [2]. As such, it has drawn unprecedented attention from the academic community [3]. ChatGPT became the fastest-growing user application in history, reaching 100 million active users as of January 2023, just two months after its launch, and recorded 14.6 billion visits in its first year [4].

ChatGPT is a powerful tool with the potential to transform medical education [4, 5]. A recent study by Aidan *et al.*, evaluated ChatGPT's potential as a medical education tool and revealed that the model achieves the equivalent of a passing score for a third-year medical student [2]. Similarly, a study by Kung *et al.* showed that ChatGPT passes the United States Medical Licensing Exam (USMLE) with a score in the 60th percentile [6]. Further, ChatGPT can be used for personalized learning, research assistance, generating case scenarios, clinical decision-making, creating content to facilitate learning, and language translation [7–10].

However, ChatGPT has introduced new challenges and threats to education [8, 10–12]. In medical education, studies have revealed various concerns including issues with academic integrity, data accuracy, potential detriments to learning, plagiarism, privacy, and security concerns [9, 13–16]. Currently, there is no reliable way to differentiate between ChatGPT written text and human-written text. This poses a great challenge to educators when marking essays. When used responsibly; ChatGPT has potential to revolutionize medical education. AI tools such as ChatGPT are valuable supplementary resources, more especially in areas where access to up-to-date medical textbooks and academic materials may be limited due to resource constraints [17]. Additionally, AI tools can address faculty shortages, support research, and

innovation, and advance critical thinking skills [17, 18]. To fully utilize these benefits, there is an urgent need to develop institutional AI policies and guidelines, thereby harnessing the advantages while mitigating associated risks in medical education [17]. Universities in some high-income countries have developed guidelines to guide the use of AI in education. However, institutions in Uganda and many other LMICs have no guidelines on how to use ChatGPT and other similar Artificial Intelligence (AI) tools [19, 20]. To develop such guidance; we need to know: 1) what proportion of medical students are using ChatGPT and other AI tools; 2) what are the common AI tools used by medical students; 3) what are these AI tools being used for; and 4) what factors are associated with the use of AI tools among medical students.

## Materials and methods

### Study design

This was a descriptive cross-sectional study aimed at assessing the use of ChatGPT and other AI tools among medical students in Uganda.

### Study setting

The study was conducted at medical schools of four public universities in Uganda from 1st November 2023 to 20th December 2023. These included Busitema University, Makerere University, Mbarara University of Science and Technology, and Gulu University. These universities were selected because they are the largest and oldest public universities that offer undergraduate medical degrees in Uganda.

Busitema University medical school was established in 2013 and is located in Mbale city, eastern Uganda. It provides medical education at undergraduate and postgraduate levels.

Makerere University Medical school was founded in 1924 and is located on Mulago Hill in north-east Kampala, Uganda's capital and largest city. It provides medical education at diploma, undergraduate, and postgraduate levels.

Mbarara University of Science and Technology was founded in 1989 and is located on the premises of Mbarara Regional Referral Hospital, in the city of Mbarara, Western Uganda.

Gulu university medical school was founded in 2004 and is located Gulu city, the largest urban center in Northern Uganda, approximately 345 kilometers (214 mi), by road, north of Kampala, Uganda's capital and largest city. It provides medical education at diploma, undergraduate and postgraduate levels.

### Study population

We enrolled undergraduate medical students pursuing Bachelor of Medicine and Bachelor of Surgery at the selected universities. Undergraduate medical students not available at the time of study were excluded.

### Sample size estimation and sampling technique

This was calculated by Open-Epi calculator (http://www.openepi.com). We assumed a 50% prevalence of Open AI tools use, precision of 5%, and design effect of 1.5. This gave us a total sample size of 639 after assuming a non-response rate of 10%. Of these, we were able to reach out to 564 participants, 75 were unreachable. Stratified random sampling was used to recruit the eligible participants. Random sampling was done in each stratum based upon the percentage that each subgroup represented in the population. A list of all the students at each faculty of the selected universities was obtained, and grouped as per the year of study. Random

numbers were assigned to the names, they were then filled in a random selector program, that generated a random list of the numbers to which the names were attached. The selected students were approached and given information about the study. Those that consented to participate in the study were recruited and the link to the questionnaire was sent to them.

### Study variables

Our outcome variable was use of AI tools such as ChatGPT among medical students. Participants were asked if they have ever used AI tools such as ChatGPT, this was recorded as Yes (denoted as 1) or No (denoted as 0). The independent variables included socio-demographic characteristics such as age, religion, marital status, sex, university, year of study and awareness about AI tools.

### Data collection tool and procedures

We used a semi structured self-administered questionnaire developed based on literature [17] to collect data. The questionnaire included questions on participant's socio-demographics (age, sex, religion, course of study and year of study) and use of AI tools. Questions under use of AI tools assessed awareness about AI tools, actual use of AI tools, which AI tools are being used and what these AI tools are being used for. Data were collected by twenty trained research assistants who were medical students in the selected universities (five students per university). Eligible participants were approached by the research assistant, informed about the study and those who consented were sent to a link to the questionnaire via their email or WhatsApp number. The questionnaire was developed electronically in KoBo Toolbox which is an open-source software developed by the Harvard Humanitarian Initiative with support from United Nations agencies, CISCO, and partners to support data management by researchers and humanitarian organizations (https://www.kobotoolbox.org/). The servers are secure and encrypted with strong safe guards and protection against data loss. participants' socio-demographics including.

### Data analysis

Data were cleaned and analyzed in Stata version 17.0 (StataCorp; College Station, TX, USA). We summarized categorical data using frequencies and percentages, and numerical data as mean (standard deviation) and median (interquartile range) as appropriate. Bar graphs were used for data visualization. We conducted a generalized linear regression for the Poisson family with a log link to estimate prevalence ratios between use of AI tools and various exposures. We used robust cluster variance estimation to adjust for clustering at the University level. Factors with a p-value less than 0.05 at multivariable regression analysis were considered significant.

### Ethical considerations

Ethics approval to conduct the study was granted by the Busitema University Research and Ethics committee (BUFHS-2023-79). Written informed consent was obtained from all the participants before recruitment into the study. Participants were informed about the study objectives, risks, and benefits before obtaining their consent and they received an internet data refund after taking part in the study. There was no coercion for participation in the study, and participants were free to withdraw from the study at any time. Identifiable information such as participant names weren't collected, a unique code was assigned to each participant to ensure anonymity. The data collected were kept in a secure computer which was accessible only to the investigators.

## Results

### Participant characteristics (Table 1)

A total of 564 medical students participated in this study. Majority of the respondents (72.0%; 406/564) were aged between 18 and 25 years. The median age (IQR) was 23.0 (22.0–26.5). More than two-thirds (70.0%; 395/564) were male and the majority (81.7%; 461/564) were single. More than a quarter (30.5%; 172/564) were Anglicans, and (32.3%; 182/564) were Catholics. A third of the students, (33.3%; 190/564) were from Mbarara University of Science and Technology, (28.5%; 161/564) Busitema University, (23.8%; 134/564) Makerere University and (14.0%; 79/564) Gulu University.

### Use of ChatGPT and other AI tools among medical students

Almost all 93% (522/564) had heard about AI tools. Most heard about AI tools from friends (71.0%), 41.6% (217/522) social media, 5.8% (30/522) lecturer and 2% (11/522) cited other sources which included; parents, and religious leader. A total of 427 out of 564 participants

**Table 1. Characteristics of the participants.**

| Characteristic (n = 564) | Frequency (%) |
|---|---|
| **Age in years** | |
| 1. 18–25 | 406 (72.0%) |
| 2. 26–34 | 131 (23.2%) |
| 3. 35–46 | 27 (4.8%) |
| **Sex** | |
| 1. Female | 169 (30.0%) |
| 2. Male | 395 (70.0%) |
| **Religion** | |
| 1. Anglican/Protestant | 172 (30.5%) |
| 2. Pentecostal/Born again | 182 (32.3%) |
| 3. Catholic | 182 (32.3%) |
| 4. Muslim | 47 (8.3%) |
| 5. Seventh Day Adventist | 34 (6.0%) |
| 6. Others* | 10 (1.8%) |
| **University** | |
| 1. Gulu University | 79 (14.0%) |
| 2. Busitema University | 161 (28.5%) |
| 3. Makerere University | 134 (23.8%) |
| 4. Mbarara University of Science and Technology | 190 (33.7%) |
| **Current year of study** | |
| 1. Year 1 | 66 (11.7%) |
| 2. Year 2 | 74 (13.1%) |
| 3. Year 3 | 167 (29.6%) |
| 4. Year 4 | 149 (26.4%) |
| 5. Year 5 | 108 (19.1%) |
| **Marital status** | |
| 1. Single | 461 (81.7%) |
| 2. Married | 103 (18.3%) |

*Jehovah's witness, atheist, traditionalist

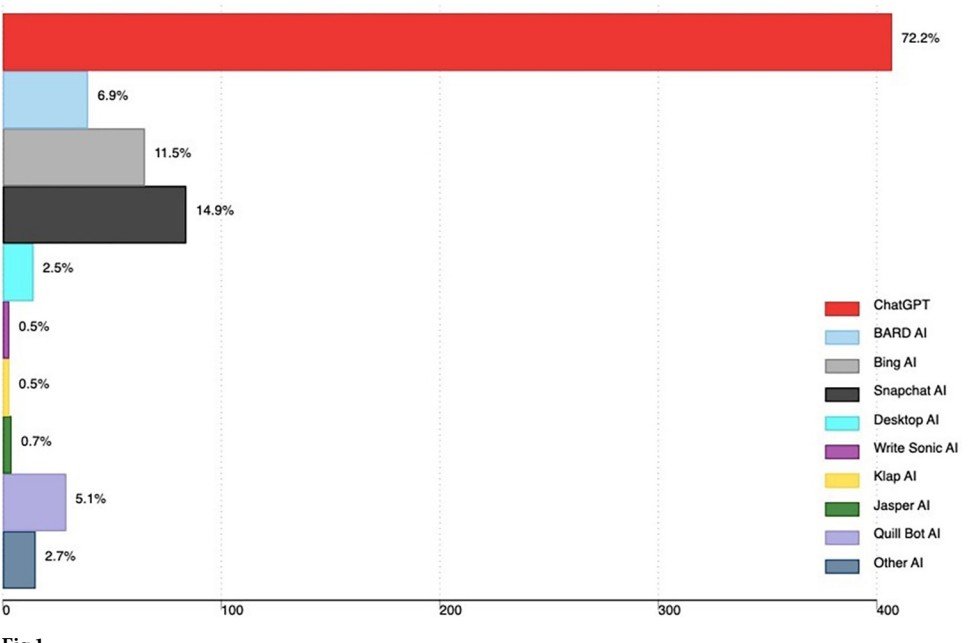

**Fig 1.**

(75.7%) had ever used AI tools. Majority 72.2% had used ChatGPT, 14.9% Snapchat AI, 11.5% Bing AI and 6.9% Bard AI. Other AI tools are in Fig 1.

Regarding areas of applications, students used AI tools for both academic and non-academic purposes. For academic purposes, most students reported that they use AI tools to complete assignments (55.5%), preparing for tutorials (39.9%), preparing for exams (34.8%) and research writing (24.8%). Other academic uses are in Fig 2.

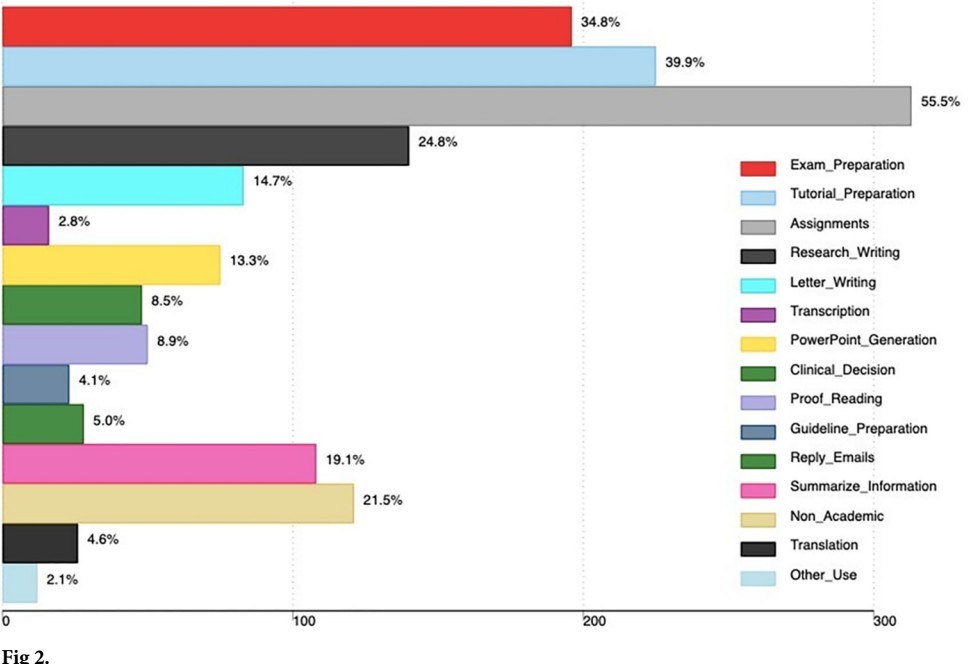

**Fig 2.**

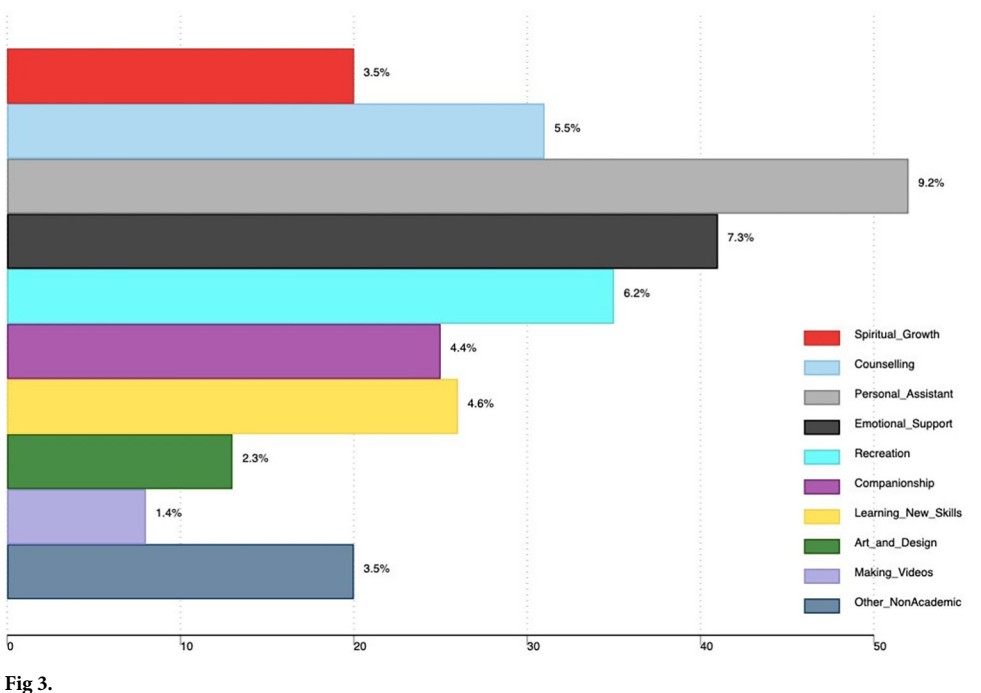

**Fig 3.**

Non-academic uses reported by the students include; AI tools as a personal assistant (9.2%), emotional support (7.3%), recreation (6.2%) and counselling (5.5%). Other non-academic uses are shown in Fig 3.

### Factors associated with use of ChatGPT and other AI tools among medical students in Uganda (Table 2)

Medical students aged 35 to 46 years were 31% less likely to use AI tools as compared to those aged less than 35 years (aPR: 0.69; 95% CI: [0.62, 0.76]).

Students in Makerere University were 66% more likely to use AI tools compared to students in Gulu University (aPR: 1.66; 95% CI: [1.64, 1.69]).

Year two medical students were 18% less likely to use AI tools as compared to those in year one (aPR: 0.82; 95% CI: [0.72, 0.94]).

### Discussion

In this study, we assessed the use of ChatGPT and other AI tools among medical students in Uganda. The use of ChatGPT and other AI tools among medical students was widespread (76%). This could be due to increased awareness about AI tools among students as shown in our findings; almost all students (93%) had ever heard about AI tools. In addition, the increased familiarity among students regarding the capabilities and benefits of AI tools in medical education could have contributed to the widespread use of AI tools. Comparable to our findings, a study in Germany found a 63.4% use of AI tools among students [21]. However, a study by Weidener et al., revealed a relatively lower AI use of 38.8% among medical students in Germany, Austria, and Switzerland [22]. Furthermore, a study in Jordan among health students showed that only 11.3% had ever used AI tools such as ChatGPT [17]. Another study in Jordan and the West Bank of Palestine among medicine and pharmacy students also showed that despite most students being aware of AI tools, less than half had used them in their

**Table 2. Factors associated with use of ChatGPT and other AI tools among medical students in Uganda.**

| Variable | cPR [95% CI] | P-value | aPR [95% CI] | P-value |
|---|---|---|---|---|
| **Age Category** | | | | |
| 18–25 | 1 | | 1 | |
| 26–34 | 0.79 [0.66, 0.95] | 0.012 | 0.87 [0.67, 1.12] | 0.280 |
| 35–46 | 0.64 [0.56, 0.73] | <0.001 | **0.69 [0.62, 0.76]** | **<0.001** |
| **Sex** | | | | |
| Female | 1 | | 1 | |
| Male | 0.96 [0.80, 1.14] | 0.614 | 1.01 [0.93, 1.08] | 0.868 |
| **Religion** | | | | |
| Anglican/Protestant | 1 | | 1 | |
| Pentecostal/Born again | 0.95 [0.88, 1.03] | 0.217 | 0.99 [0.91, 1.08] | 0.870 |
| Catholic | 0.94 [0.82, 1.08] | 0.371 | 0.95 [0.85, 1.05] | 0.285 |
| Muslim | 0.87 [0.62, 1.21] | 0.397 | 0.85 [0.61, 1.19] | 0.344 |
| Seventh Day Adventist | 1.05 [0.93, 1.18] | 0.437 | 1.02 [0.93, 1.12] | 0.647 |
| Others | 1.15 [1.04, 1.27] | 0.009 | 1.13 [1.02, 1.26] | 0.016 |
| **University** | | | | |
| Gulu University | 1 | | 1 | |
| Busitema University | 1.22 [1.22, 1.22] | <0.001 | **1.31 [1.21, 1.41]** | **<0.001** |
| Makerere University | 1.70 [1.70, 1.70] | <0.001 | **1.66 [1.64, 1.69]** | **<0.001** |
| Mbarara University of Science and Technology | 1.58 [1.58, 1.58] | <0.001 | **1.58 [1.54, 1.62]** | **<0.001** |
| **Year of Study** | | | | |
| Year 1 | 1 | | 1 | |
| Year 2 | 0.71 [0.54, 0.94] | 0.017 | **0.82 [0.72, 0.94]** | **0.005** |
| Year 3 | 0.80 [0.71, 0.89] | <0.001 | 0.95 [0.83, 1.09] | 0.467 |
| Year 4 | 0.83 [0.59, 1.16] | 0.276 | 1.03 [0.91, 1.17] | 0.665 |
| Year 5 | 0.88 [0.71, 1.08] | 0.214 | 0.96 [0.88, 1.04] | 0.339 |

education [23]. The discrepancy across studies could be explained by the differences in study periods and characteristics of the study populations. For instance, the study in Jordan was done between February and March 2023 [17], probably AI tools had not gained popularity among students during that time as compared to our study done between November and December 2023, where AI tools had gained much popularity amongst the students. Additionally, some AI tools weren't accessible before mid-2023, for instance, Bing Chat was not broadly accessible until May 2023 [24].

Our findings showed that ChatGPT was the most commonly (72%) used AI tool among medical students. Similarly, studies done in Germany and Nigeria revealed that the majority of the students indicated ChatGPT as the tool they use [21, 25]. This could be attributed to the fact that ChatGPT is the first generative AI tool readily available to the public and the fact that it is easy to use [12]. Furthermore, ChatGPT's performance in answering various medical exams has been evaluated and it has demonstrated the potential to pass most of these exams [5, 6, 26]. Other AI tools used by more than 5% of the students included Snapchat AI, Bing AI, and Bard AI. Most of these AI tools became freely available to the public in mid-2023, probably they hadn't gained much popularity among students as compared to ChatGPT which was made freely accessible to the public on November 30, 2022 [24]. However, studies conducted in Germany and India also revealed the use of Bing and Bard AI tools among medical students [17, 21].

Our study revealed that students used AI tools to complete assignments, to prepare for tutorials, and to prepare for exams. Consistent with our findings, several studies have revealed that

students use AI tools to complete assignments and possibly cheat in exams [16, 21, 27]. AI tools such as ChatGPT can generate human-like texts to user questions and thus can be used by the students to complete written assignments and online examinations [8]. Students also reported the use of AI tools for research writing, summarizing information, translation, transcription, creating power point presentations, and clinical decision making. Similar findings were revealed in a study by Biri *et al.*, among medical students in India [17]. This indicates a growing recognition of AI tools as applications that can bridge gaps in understanding and enhancing medical students' learning experience [17].

We also found that students use AI tools for non-academic purposes including; personal assistants, emotional support, counseling, spiritual growth, recreation, companionship, and learning new skills. Further studies need to explore the role of AI tools in the mental and social health of medical students.

Medical students aged greater than 35 years were 31% less likely to use AI tools as compared to those aged less than 35 years. In Uganda, there are undergraduate medical students who are aged more than 35 years. These majorly include those who study medical course at diploma level and later join medical school for a medical undergraduate bachelor degree. Older students may be discouraged from using AI due to fixed beliefs that they are not efficient with technology [28–30]. Furthermore, creators do not often consider the older generation when designing novel technology, thus biased perspectives on AI use emerge. Additionally, older adults are slower to adopt new technology compared to younger adults [28–30].

Our findings also revealed that students at Makerere University were 66% more likely to use AI tools compared to students at Gulu University. Students at Makerere University could have a higher exposure to AI as compared to those at Gulu University. Makerere University has high student numbers as compared to Gulu University and is located in the capital city of Uganda, an area more urban than Gulu city where Gulu University is located. Urban areas promote technology awareness and use. In our study, year two medical students were 18% less likely to use AI tools than those in year one. Students in year one could be more exposed to AI as compared to those in year two. Further, year one students joined the university in August 2023, when AI tools such as ChatGPT had just become famous at our universities and thus are more likely to use AI than year two students who were already used to the conventional methods of learning.

Findings from this study do contribute to the emerging debates around the use of AI tools by medical students and have key implications for learning. Educators should assume students are using AI and adjust their way of teaching and setting of exams to suit this new reality. Further, educators and institutions need to develop AI policies and guidelines to ensure that future medical professionals are adequately prepared to navigate the challenges and opportunities presented by AI in medical education. Universities in high-income countries have come up with guidelines including *Guidelines for the Use of Artificial Intelligence in University Courses* by Juan David Gutiérrez at Universidad del Rosario, *Initial guidelines for the use of Generative AI tools at Harvard* by Havard University, *A Guide for Students*: *Should I use AI*? by Ulster University and *Student guidance on using Generative Artificial Intelligence tools ethically for study* by the University of Birmingham [19, 20, 31, 32]. The guidelines focus on the need to promote AI literacy, the need to cite AI sources, and the limitations of AI [19, 20, 33]. The guidelines also emphasize informed, transparent, ethical and responsible use of high-risk AI (generative AI such as Chat-GPT and stable diffusion AI such as DALL-E 2) in education [31]. Informed use requires that prior to using the tool, the student should research who or what company developed the tool, how it was developed, how it works, what functions it can perform, and what limitations and/or risks it presents [31]. Transparent use entails indicating in detail which tool the student used and how he/she used it [31]. Ethical use includes ensuring

that one must distinguish what was written or produced directly by the student and what was generated by an AI tool [31]. Responsible use emphasizes that the use of these AI tools should be limited to early stages of the student's work, to inspire or suggest directions, not to produce content that will later be included in his/her deliverables [31]. Guidance has also included examples of what AI can do well and what AI cannot do well [20]. Universities in low-income countries could borrow upon these guidelines and contextualize them. The guidelines would define the boundaries in which AI should be used in education. A key difference between high-income countries is the fact that most students in low-income countries use unpaid versions that are not up to date [34]. As such, guidelines in low-income countries should contextualize AI use [35].

## Strengths and limitations

To the best of our knowledge, this is one of the few studies done to assess the use of AI tools such as ChatGPT among medical students in low-resource settings. In addition, we included four universities, one from each region of the country thus our findings may be generalizable to all medical students in the country. Although our findings provide valuable insights into the state of AI use in medical education in Uganda, broader multinational studies would offer a more comprehensive understanding. One key limitation of our study is the reliance on self-reported data from medical students, which might be subject to social desirability bias. To mitigate this, we used a self-administered questionnaire for data collection, the questionnaires were anonymous and the students were given identification numbers. As such the social pressure while answering questions was reduced.

## Conclusion

The use of ChatGPT and other AI tools was widespread among medical students in Uganda. AI tools were used for both academic and non-academic purposes. Younger students were more likely to use AI tools compared to older students. There is a need for AI training programs in institutions to empower older students with essential skills for the digital age. Further, our research adds more evidence to existing voices calling for regulatory frameworks of AI in medical education to ensure that future medical professionals are adequately prepared to navigate the challenges and opportunities presented by AI in medical education.

## Supporting information

**S1 Dataset. Anonymised data set.**
(XLSX)

## Acknowledgments

The authors would like to thank the research assistants, study participants, university staff, and members of the HEPI consortium.

## Author Contributions

**Conceptualization:** Elizabeth Ajalo, David Mukunya, Ritah Nantale, Joseph Luwaga Mpagi, Milton W. Musaba, Faith Oguttu, Job Kuteesa, Aloysius Gonzaga Mubuuke, Ian Guyton Munabi, Sarah Kiguli.

**Data curation:** Elizabeth Ajalo, David Mukunya, Ritah Nantale, Joseph Luwaga Mpagi, Milton W. Musaba, Faith Oguttu, Job Kuteesa, Aloysius Gonzaga Mubuuke, Ian Guyton Munabi, Sarah Kiguli.

**Formal analysis:** David Mukunya, Ritah Nantale, Frank Kayemba, Kennedy Pangholi, Yakobo Baddokwaya Nsubuga, Faith Oguttu, Aloysius Gonzaga Mubuuke, Ian Guyton Munabi, Sarah Kiguli.

**Funding acquisition:** David Mukunya, Sarah Kiguli.

**Investigation:** Elizabeth Ajalo, Frank Kayemba, Kennedy Pangholi, Jonathan Babuya, Suzan Langoya Akuu, Amelia Margaret Namiiro, Yakobo Baddokwaya Nsubuga, Milton W. Musaba.

**Methodology:** Elizabeth Ajalo, David Mukunya, Ritah Nantale, Frank Kayemba, Kennedy Pangholi, Jonathan Babuya, Suzan Langoya Akuu, Amelia Margaret Namiiro, Yakobo Baddokwaya Nsubuga, Milton W. Musaba, Faith Oguttu, Job Kuteesa, Aloysius Gonzaga Mubuuke, Ian Guyton Munabi.

**Project administration:** Elizabeth Ajalo, David Mukunya, Joseph Luwaga Mpagi, Milton W. Musaba, Aloysius Gonzaga Mubuuke, Sarah Kiguli.

**Software:** David Mukunya.

**Supervision:** Ritah Nantale, Amelia Margaret Namiiro, Joseph Luwaga Mpagi.

**Validation:** Ritah Nantale, Frank Kayemba, Kennedy Pangholi, Jonathan Babuya, Suzan Langoya Akuu, Amelia Margaret Namiiro, Joseph Luwaga Mpagi, Milton W. Musaba.

**Visualization:** David Mukunya, Ritah Nantale, Frank Kayemba, Kennedy Pangholi, Jonathan Babuya, Amelia Margaret Namiiro.

**Writing – original draft:** Elizabeth Ajalo, David Mukunya, Ritah Nantale, Frank Kayemba, Jonathan Babuya, Amelia Margaret Namiiro, Joseph Luwaga Mpagi, Milton W. Musaba, Faith Oguttu, Job Kuteesa, Aloysius Gonzaga Mubuuke, Ian Guyton Munabi, Sarah Kiguli.

**Writing – review & editing:** Elizabeth Ajalo, David Mukunya, Ritah Nantale, Frank Kayemba, Kennedy Pangholi, Suzan Langoya Akuu, Yakobo Baddokwaya Nsubuga, Joseph Luwaga Mpagi, Milton W. Musaba, Faith Oguttu, Job Kuteesa, Aloysius Gonzaga Mubuuke, Ian Guyton Munabi, Sarah Kiguli.

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
