## [Decision Letter · Decision Letter 0]

24 Mar 2024

PONE-D-24-01790Widespread use of ChatGPT and other Artificial Intelligence tools among medical students in Uganda: a cross-sectional studyPLOS ONE

Dear Dr. Nantale,

Thank you for submitting your manuscript to PLOS ONE. After careful consideration, we feel that it has merit but does not fully meet PLOS ONE’s publication criteria as it currently stands. Therefore, we invite you to submit a revised version of the manuscript that addresses the points raised during the review process.

**ACADEMIC EDITOR:**In addition to the reviewers' comments,1. Data visualization could be improved. Could we present some of the figures using charts other than bar graphs?. For bar charts, please indicate the axis titles.2. Please expand the DISCUSSION. There are new reports on this aspect in other countries, which may make your discussion robust, and probably lead to more meaningful conclusions. Weidener L, Fischer M. Artificial Intelligence in Medicine: Cross-Sectional Study Among Medical Students on Application, Education, and Ethical Aspects. JMIR Med Educ 2024;10:e51247https://mededu.jmir.org/2024/1/e51247Kapsali M, Livanis E, Tsalikidis C, Oikonomou P, Voultsos P, Tsaroucha A. Ethical Concerns About ChatGPT in Healthcare: A Useful Tool or the Tombstone of Original and Reflective Thinking?. Cureus 2024, 16(2): e54759. doi:10.7759/cureus.54759Heredia-Negrón, F.; Tosado-Rodríguez, E.L.; Meléndez-Berrios, J.; Nieves, B.; Amaya-Ardila, C.P.; Roche-Lima, A. Assessing the Impact of AI Education on Hispanic Healthcare Professionals’ Perceptions and Knowledge. *Educ. Sci.* 2024, *14*, 339. https://doi.org/10.3390/educsci14040339McLennan S, Meyer A, Schreyer K, Buyx A (2022) German medical students´ views regarding artificial intelligence in medicine: A cross-sectional survey. PLOS Digit Health 1(10): e0000114. https://doi.org/10.1371/journal.pdig.00001143. It is necessary to indicate the potential limitations of the study, just as you indicated the strengths.  4. Your conclusions should be expanded in the broader context, and should preferably give some directions for future reseach.

We look forward to receiving your revised manuscript.

Kind regards,

Timothy Omara

Academic Editor

PLOS ONE

Journal Requirements:

3. We note you have included a table to which you do not refer in the text of your manuscript. Please ensure that you refer to Table 1 and 2 in your text; if accepted, production will need this reference to link the reader to the Table.

Reviewers' comments:

Reviewer's Responses to Questions

**Comments to the Author**

1. Is the manuscript technically sound, and do the data support the conclusions?

Reviewer #1: Yes

Reviewer #2: Yes

2. Has the statistical analysis been performed appropriately and rigorously? 

Reviewer #1: Yes

Reviewer #2: Yes

3. Have the authors made all data underlying the findings in their manuscript fully available?

Reviewer #1: Yes

Reviewer #2: Yes

4. Is the manuscript presented in an intelligible fashion and written in standard English?

Reviewer #1: Yes

Reviewer #2: No

5. Review Comments to the Author

Reviewer #1: I am pleased to give my thoughts on the article titled 'Widespread Use of ChatGPT and Other Artificial Intelligence Tools Among Medical Students in Uganda: A Cross-Sectional Study.' Overall, it appears to be a promising article of good quality. However, it can be improved.

Title: The title could be improved removing redundant words by specifying the type of AI tools examined in the study. For example, "Widespread use of ChatGPT and Other Artificial Intelligence Tools among Medical Students in Uganda: A Cross-Sectional Study."

Abstract: Consider including specific findings or percentages in the abstract to give readers a clearer understanding of the key results. Also, mentioning the implications of the findings in the abstract would enhance its significance.

Introduction: The introduction effectively sets the context by explaining the significance of AI tools like ChatGPT in medical education. However, it could be strengthened by providing more context on the challenges faced by medical education due to the widespread adoption of AI tools. Consider integrating more recent literature or studies to emphasize the relevance and timeliness of the research.

Materials and Methods: Provide more details on the semi-structured questionnaire used, including the specific questions related to AI tool usage. Describe the process of questionnaire adaptation and validation, if applicable, to ensure the reliability of data collection instruments.

Mention the steps taken to ensure the confidentiality and anonymity of participants' responses.

Results: The results section presents the key findings of the study in a structured manner.

Discussion: The discussion effectively contextualizes the findings within existing literature and highlights their implications. However, the authors should provide a more detailed comparison of the study findings with similar research conducted in other settings or populations.

While the authors discuss the strength of the, they should consider discussing potential limitations of the study and their impact on the interpretation of results. Also, propose recommendations for future research or practical interventions based on the study findings.

Conclusion: Consider reiterating the implications of the study findings for medical education policy and practice. Provide clear recommendations for addressing the challenges and opportunities associated with the widespread use of AI tools among medical students. Avoid introducing new information or findings in the conclusion section.

Ethical Approval and Consent to Participate: Consider including information on how participants were informed about the study objectives, risks, and benefits before obtaining their consent. Specify whether any incentives or compensation were provided to participants for their involvement in the study.

General Comments: The article would benefit from thorough proofreading to correct grammatical errors, typos, and formatting inconsistencies.

Reviewer #2: 1. “In order to exploit the use of ChatGPT and other related AI software, there is an urgent need to develop guidelines.” – do the authors mean to say, in order to prevent exploitation?

2. What were the measures taken to reduce social desirability bias in the study? Will students actually acknowledge its use for plagiarism?

3. Are there any guidelines in developed countries / HMICs on how to use AI tools?

4. How can guidelines be formed to limit its misuse? – Will universities / journals check for AI written content?

5. “Almost all 93% (522/564) had ever had about AI tools 155 such as ChatGPT.” – Needs to be rephrased

6. “Most 71%” – most and percentage should be avoided together

7. Results need to be rewritten in a better language

8. “Medical students aged 35 to 46 years were 31% less likely to use AI tools as compared to those aged less than 35 years” – Are undergraduate medical students aged above 35, this is highly unusual across the world? Also can 13 really be compared against 47 and 77 to draw such a definite conclusion?

9. While the study provides insights on the patterns on Generative AI use, and the need for regulatory guidelines – Can the authors discuss steps taken in other countries to reduce its misuse.

6. PLOS authors have the option to publish the peer review history of their article (what does this mean?). If published, this will include your full peer review and any attached files.

Reviewer #1: **Yes: **Dr. Amir Kabunga

Reviewer #2: **Yes: **Rajmohan Seetharaman

---

## [Author Response · Author response to Decision Letter 0]

14 Aug 2024

Reviewer’s comment Response to comment Line number

Academic Editor

1. Data visualization could be improved. Could we present some of the figures using charts other than bar graphs?. For bar charts, please indicate the axis titles. Thank you for this comment. We have explored the use of pie-charts, donut charts, lollipop charts and upset plots without much success. The strength of the current chart is that it shows both absolute numbers on the x-axis and the corresponding percentage in brackets. We kindly request that you suggest the better chart to use. N/A

2. Please expand the DISCUSSION. There are new reports on this aspect in other countries, which may make your discussion robust, and probably lead to more meaningful conclusions. Thank you, we have revised the discussion basing on findings from new articles on Artificial Intelligence in medical education. 

3. It is necessary to indicate the potential limitations of the study, just as you indicated the strengths. Thank you, we have described the limitations of our study.

“One key limitation of our study is the reliance on self-reported data from medical students, which might be subject to social desirability bias. However, to reduce social desirability bias, we used a self-administered questionnaire for data collection” 280-282

4. Your conclusions should be expanded in the broader context, and should preferably give some directions for future research.

 Thank you, this has been done. 284-290

Reviewer #1

Title: The title could be improved removing redundant words by specifying the type of AI tools examined in the study. For example, "Widespread use of ChatGPT and Other Artificial Intelligence Tools among Medical Students in Uganda: A Cross-Sectional Study." Thank you for your comment. We have adjusted our title accordingly. However, the other AI tools used are many and shall make the title very lengthy and have not included them. N/A

Abstract: Consider including specific findings or percentages in the abstract to give readers a clearer understanding of the key results. Also, mentioning the implications of the findings in the abstract would enhance its significance. Thank you, this has been done. 40-55

Introduction: The introduction effectively sets the context by explaining the significance of AI tools like ChatGPT in medical education. However, it could be strengthened by providing more context on the challenges faced by medical education due to the widespread adoption of AI tools. Consider integrating more recent literature or studies to emphasize the relevance and timeliness of the research. Thank you, this has been done. 69-88

Materials and Methods: Provide more details on the semi-structured questionnaire used, including the specific questions related to AI tool usage. Describe the process of questionnaire adaptation and validation, if applicable, to ensure the reliability of data collection instruments. Thank you, this has been done and we have also uploaded the questionnaire we used. 136-140

Mention the steps taken to ensure the confidentiality and anonymity of participants' responses. Thank you, this has been done 298-301

Discussion: The discussion effectively contextualizes the findings within existing literature and highlights their implications. However, the authors should provide a more detailed comparison of the study findings with similar research conducted in other settings or populations. We have revised the discussion accordingly. 206-271

While the authors discuss the strength of the, they should consider discussing potential limitations of the study and their impact on the interpretation of results. Also, propose recommendations for future research or practical interventions based on the study findings. Thank you, we have described the limitations of our study.

“One key limitation of our study is the reliance on self-reported data from medical students, which might be subject to social desirability bias. However, to reduce social desirability bias, we used a self-administered questionnaire for data collection” 278-281

Conclusion: Consider reiterating the implications of the study findings for medical education policy and practice. Provide clear recommendations for addressing the challenges and opportunities associated with the widespread use of AI tools among medical students. Avoid introducing new information or findings in the conclusion section. Thank you, we have revised accordingly.

 283-289

Ethical Approval and Consent to Participate: Consider including information on how participants were informed about the study objectives, risks, and benefits before obtaining their consent. Specify whether any incentives or compensation were provided to participants for their involvement in the study. Thank you for the comment. This has been done.

 293-300

General Comments: The article would benefit from thorough proofreading to correct grammatical errors, typos, and formatting inconsistencies. Thank you, we have done a proofreading and corrected the identified grammatical errors, and typos N/A

Reviewer #2 

1. “In order to exploit the use of ChatGPT and other related AI software, there is an urgent need to develop guidelines.” – do the authors mean to say, in order to prevent exploitation? Thank you for your comment. With exploiting, we refer to making use of. We have rephrased it to “AI tools can address faculty shortages, support research, and innovation, and advance critical thinking skills. In order to fully exploit these benefits, there is an urgent need to develop institutional AI policies and guidelines, thereby harnessing the advantages while mitigating associated risks in medical education” 

2. What were the measures taken to reduce social desirability bias in the study? Will students actually acknowledge its use for plagiarism? To reduce social desirability bias, we used a self-administered questionnaire. 136

3. Are there any guidelines in developed countries / HMICs on how to use AI tools?

 Yes, there are guidelines in some high-income countries on AI use in education N/A

4. How can guidelines be formed to limit its misuse? – Will universities / journals check for AI written content? There are attempts to check for AI written content. However, most of these tools are not yet validated. 

5. “Almost all 93% (522/564) had ever had about AI tools 155 such as ChatGPT.” – Needs to be rephrased Thank you, this has been rephrased. 162

6. “Most 71%” – most and percentage should be avoided together Thank you, this has been rephrased. 163

7. Results need to be rewritten in a better language Thank you, this has been done. 158-204

8. “Medical students aged 35 to 46 years were 31% less likely to use AI tools as compared to those aged less than 35 years” – Are undergraduate medical students aged above 35, this is highly unusual across the world? Also can 13 really be compared against 47 and 77 to draw such a definite conclusion? We understand your concern. However, in Uganda, there are undergraduate medical students who are aged more than 35 years. These majorly include those who study medical course at diploma level and later join medical school for a medical undergraduate bachelor degree. N/A

9. While the study provides insights on the patterns on Generative AI use, and the need for regulatory guidelines – Can the authors discuss steps taken in other countries to reduce its misuse. Thank you, we have discussed these as indicated. Other countries have developed AI policies and guidelines to prevent AI misuse. 274-280

---

## [Decision Letter · Decision Letter 1]

26 Aug 2024

PONE-D-24-01790R1Widespread use of ChatGPT and other Artificial Intelligence tools among medical students in Uganda: a cross-sectional studyPLOS ONE

Dear Dr. Nantale,

Thank you for submitting your manuscript to PLOS ONE. After careful consideration, we feel that it has merit but does not fully meet PLOS ONE’s publication criteria as it currently stands. Therefore, we invite you to submit a revised version of the manuscript that addresses the points raised during the review process.

 Please submit your revised manuscript by Oct 10 2024 11:59PM. If you will need more time than this to complete your revisions, please reply to this message or contact the journal office at plosone@plos.org. Please include the following items when submitting your revised manuscript:A rebuttal letter that responds to each point raised by the academic editor and reviewer(s). You should upload this letter as a separate file labeled 'Response to Reviewers'.A marked-up copy of your manuscript that highlights changes made to the original version. You should upload this as a separate file labeled 'Revised Manuscript with Track Changes'.An unmarked version of your revised paper without tracked changes. You should upload this as a separate file labeled 'Manuscript'.

We look forward to receiving your revised manuscript.

Kind regards,

Timothy Omara, PhD

Academic Editor

PLOS ONE

Additional Editor Comments:

Dear authors,

The reviewers have re-assessed your resubmission. However, there are still suggestions that need to be incorporated. In addition to reviewer comments, I suggest taking a closer look at the following aspects of the draft.

1. Table 1 seems not to be properly reported. I would expect that it gives a general overview of the overall sociodemographic characteristics of the respondents (irrespective of whether or not they have ever used ChatGPT or any other such LLM). Please refer to my suggestions in the manuscript file attached.

2. It is also evident that percentage calculations in Figures 1, 2 and 3 are erroneous. For example, the total percentage in Figure 1 is 118% instead of 100%. Figure 3 has total percentage less than 100%. It would be expected to group academic and non-academic uses of these LLM together, because this is the collective sum of what students attested to using them for.

3. After these revisions, you may have to redo statistical analysis and change some parts of the abstract, results, discussion and conclusions.

Reviewers' comments:

Reviewer's Responses to Questions

**Comments to the Author**

1. If the authors have adequately addressed your comments raised in a previous round of review and you feel that this manuscript is now acceptable for publication, you may indicate that here to bypass the “Comments to the Author” section, enter your conflict of interest statement in the “Confidential to Editor” section, and submit your "Accept" recommendation.

Reviewer #2: (No Response)

2. Is the manuscript technically sound, and do the data support the conclusions?

Reviewer #2: Partly

3. Has the statistical analysis been performed appropriately and rigorously? 

Reviewer #2: Yes

4. Have the authors made all data underlying the findings in their manuscript fully available?

Reviewer #2: Yes

5. Is the manuscript presented in an intelligible fashion and written in standard English?

Reviewer #2: No

6. Review Comments to the Author

Reviewer #2: Thank you for the opportunity to re-review this article. Unfortunately, many of the comments previously provided to the authors have not been adequately addressed. The authors should carefully reconsider each of my comments and provide more thorough responses, as the current revisions appear to be only superficial attempts to resolve the issues with the manuscript.

1. “In order to exploit the use of ChatGPT and other related AI software, there is an urgent need to develop guidelines.” – do the authors mean to say, in order to prevent exploitation?Thank you for your comment. With exploiting, we refer to making use of. We have rephrased it to “AI tools can address faculty shortages, support research, and innovation, and advance critical thinking skills. In order to fully exploit these benefits, there is an urgent need to develop institutional AI policies and guidelines, thereby harnessing the advantages while mitigating associated risks in medical education”

Reviewers comment on the response: Please replace “exploit” word with a synonym.

2. What were the measures taken to reduce social desirability bias in the study? Will students actually acknowledge its use for plagiarism?To reduce social desirability bias, we used a self-administered questionnaire.136

Reviewers comment on the response: Please explain how can self-administered questionnaire reduce social desirability bias? E.g. “Anonymity, reduced social pressure, privacy, lack of non-verbal cues, and standardized responses in self-administered questionnaires help minimize social desirability bias, leading to more accurate and reliable data collection.”

3. Are there any guidelines in developed countries / HMICs on how to use AI tools? Yes, there are guidelines in some high-income countries on AI use in educationN/A

Reviewers comment on the response: If yes then this should be mentioned and cited in the manuscript including name of the guidelines.

4. How can guidelines be formed to limit its misuse? – Will universities / journals check for AI written content?There are attempts to check for AI written content. However, most of these tools are not yet validated.

Reviewers comment on the response: Give examples and cite in the manuscript.

5. “Almost all 93% (522/564) had ever had about AI tools 155 such as ChatGPT.” – Needs to be rephrasedThank you, this has been rephrased.162

Reviewers comment on the response: Rephrasing language not up to the mark.

6. “Most 71%” – most and percentage should be avoided togetherThank you, this has been rephrased.163

Reviewers comment on the response: This is still there in the abstract “most (72.2%) had ever used ChatGPT”

7. Results need to be rewritten in a better languageThank you, this has been done.158-204

Reviewers comment on the response: This has been adequately addressed

8. “Medical students aged 35 to 46 years were 31% less likely to use AI tools as compared to those aged less than 35 years” – Are undergraduate medical students aged above 35, this is highly unusual across the world? Also can 13 really be compared against 47 and 77 to draw such a definite conclusion?We understand your concern. However, in Uganda, there are undergraduate medical students who are aged more than 35 years. These majorly include those who study medical course at diploma level and later join medical school for a medical undergraduate bachelor degree.N/A

Reviewers comment on the response: This needs to be explained in the discussion, to avoid conflicting remarks by readers.

9. While the study provides insights on the patterns on Generative AI use, and the need for regulatory guidelines – Can the authors discuss steps taken in other countries to reduce its misuse.Thank you, we have discussed these as indicated. Other countries have developed AI policies and guidelines to prevent AI misuse.274-280

Reviewers comment on the response: This discussion is very superficial. It needs a very indepth discussion – examples of universities / guideline names also need to be cited. The authors need to mention about the indepth policies.

7. PLOS authors have the option to publish the peer review history of their article (what does this mean?). If published, this will include your full peer review and any attached files.

Reviewer #2: **Yes: **Rajmohan Seetharaman

---

## [Author Response · Author response to Decision Letter 1]

7 Oct 2024

Thank you for reviewing our manuscript “PONE-D-24-01790R1; Widespread use of ChatGPT and other Artificial Intelligence tools among medical students in Uganda: a cross-sectional study”

Below is our point-by-point response to each comment.

Thank you!

Reviewer’s comment Response Line number

Editor 

1. Table 1 seems not to be properly reported. I would expect that it gives a general overview of the overall sociodemographic characteristics of the respondents (irrespective of whether or not they have ever used ChatGPT or any other such LLM). Please refer to my suggestions in the manuscript file attached. Thank you for the suggestion. We have revised the table as suggested. 

2. It is also evident that percentage calculations in Figures 1, 2 and 3 are erroneous. For example, the total percentage in Figure 1 is 118% instead of 100%. Figure 3 has total percentage less than 100%. It would be expected to group academic and non-academic uses of these LLM together, because this is the collective sum of what students attested to using them for. Thank you for your comment, for figure 1, 2 and 3, the percentages aren’t equal to 100 because questions on uses of LLM were multiple choice questions. One person could select more than one use. 

3. After these revisions, you may have to redo statistical analysis and change some parts of the abstract, results, discussion and conclusions. 

Reviewer #2 

1. “In order to exploit the use of ChatGPT and other related AI software, there is an urgent need to develop guidelines.” – do the authors mean to say, in order to prevent exploitation? Thank you for your comment. With exploiting, we refer to making use of. We have rephrased it to “AI tools can address faculty shortages, support research, and innovation, and advance critical thinking skills. In order to fully exploit these benefits, there is an urgent need to develop institutional AI policies and guidelines, thereby harnessing the advantages while mitigating associated risks in medical education”

Reviewers comment on the response: Please replace “exploit” word with a synonym. We have replaced exploit with a synonym; utilize 85

2. What were the measures taken to reduce social desirability bias in the study? Will students actually acknowledge its use for plagiarism? To reduce social desirability bias, we used a self-administered questionnaire.136

Reviewers comment on the response: Please explain how can self-administered questionnaire reduce social desirability bias? E.g. “Anonymity, reduced social pressure, privacy, lack of non-verbal cues, and standardized responses in self-administered questionnaires help minimize social desirability bias, leading to more accurate and reliable data collection.” Thank you, we used a self-administered questionnaire for data collection, the questionnaires were anonymous and the students were given identification numbers as such the social pressure while reducing questions was reduced. 

3. Are there any guidelines in developed countries / HMICs on how to use AI tools? Yes, there are guidelines in some high-income countries on AI use in education N/A

Reviewers comment on the response: If yes then this should be mentioned and cited in the manuscript including name of the guidelines. Yes, this has been updated.

“Universities in high-income countries have come up with guidelines including Guidelines for the Use of Artificial Intelligence in University Courses by Juan David Gutiérrez at Universidad del Rosario, Initial guidelines for the use of Generative AI tools at Harvard by Havard University, A Guide for Students: Should I use AI? by Ulster University and Student guidance on using Generative Artificial Intelligence tools ethically for study by the University of Birmingham” 280-284

4. How can guidelines be formed to limit its misuse? – Will universities / journals check for AI written content? There are attempts to check for AI written content. However, most of these tools are not yet validated.

Reviewers comment on the response: Give examples and cite in the manuscript. This has been done.

The guidelines focus on the need to promote AI literacy, the need to cite AI sources, and the limitations of AI. The guidelines also emphasize informed, transparent, ethical and responsible use of high-risk AI (generative AI such as Chat-GPT and stable diffusion AI such as DALL-E 2) in education. 284-296

5. “Almost all 93% (522/564) had ever had about AI tools 155 such as ChatGPT.” – Needs to be rephrased Thank you, this has been rephrased.162

Reviewers comment on the response: Rephrasing language not up to the mark. Thank you for the observation, this has been rephrased and now reads.

Almost all 93% (522/564) had ever heard about AI tools 155 such as ChatGPT 172

6. “Most 71%” – most and percentage should be avoided together. Thank you, this has been rephrased.163

Reviewers comment on the response: This is still there in the abstract “most (72.2%) had ever used ChatGPT” This has been revised as suggested. 41

7. Results need to be rewritten in a better language Thank you, this has been done.158-204

Reviewers comment on the response: This has been adequately addressed Thank you. 

8. “Medical students aged 35 to 46 years were 31% less likely to use AI tools as compared to those aged less than 35 years” – Are undergraduate medical students aged above 35, this is highly unusual across the world? Also can 13 really be compared against 47 and 77 to draw such a definite conclusion? We understand your concern. However, in Uganda, there are undergraduate medical students who are aged more than 35 years. These majorly include those who study medical course at diploma level and later join medical school for a medical undergraduate bachelor degree. N/A

Reviewers comment on the response: This needs to be explained in the discussion, to avoid conflicting remarks by readers. Thank you for the suggestion, we have explained this in the discussion.

“In Uganda, there are undergraduate medical students who are aged more than 35 years. These majorly include those who study medical course at diploma level and later join medical school for a medical undergraduate bachelor degree” 

9. While the study provides insights on the patterns on Generative AI use, and the need for regulatory guidelines – Can the authors discuss steps taken in other countries to reduce its misuse.Thank you, we have discussed these as indicated. Other countries have developed AI policies and guidelines to prevent AI misuse.274-280

Reviewers comment on the response: This discussion is very superficial. It needs a very indepth discussion – examples of universities / guideline names also need to be cited. The authors need to mention about the indepth policies. We have enriched the discussion as suggested.

“Universities in high-income countries have come up with guidelines including Guidelines for the Use of Artificial Intelligence in University Courses by Juan David Gutiérrez at Universidad del Rosario, Initial guidelines for the use of Generative AI tools at Harvard by Havard University, A Guide for Students: Should I use AI? by Ulster University and Student guidance on using Generative Artificial Intelligence tools ethically for study by the University of Birmingham

The guidelines focus on the need to promote AI literacy, the need to cite AI sources, and the limitations of AI. The guidelines also emphasize informed, transparent, ethical and responsible use of high-risk AI (generative AI such as Chat-GPT and stable diffusion AI such as DALL-E 2) in education. Informed use requires that prior to using the tool, the student should research who or what company developed the tool, how it was developed, how it works, what functions it can perform, and what limitations and/or risks it presents. Transparent use includes indicating in detail which tool the student used and how he/she used it. Ethical use includes ensuring that one must distinguish what was written or produced directly by the student and what was generated by an AI tool. Responsible use emphasizes that the use of these AI tools should be limited to early stages of the student’s work, to inspire or suggest directions, not to produce content that will later be included in his/her deliverables. Guidance has also included examples of what AI can do well and what AI cannot do well.” 284-296

---

## [Decision Letter · Decision Letter 2]

31 Oct 2024

Widespread use of ChatGPT and other Artificial Intelligence tools among medical students in Uganda: a cross-sectional study

PONE-D-24-01790R2

Dear Dr. Nantale,

We’re pleased to inform you that your manuscript has been judged scientifically suitable for publication and will be formally accepted for publication once it meets all outstanding technical requirements.

Kind regards,

Timothy Omara

Academic Editor

PLOS ONE

Additional Editor Comments (optional):

Reviewers' comments:

Reviewer's Responses to Questions

**Comments to the Author**

1. If the authors have adequately addressed your comments raised in a previous round of review and you feel that this manuscript is now acceptable for publication, you may indicate that here to bypass the “Comments to the Author” section, enter your conflict of interest statement in the “Confidential to Editor” section, and submit your "Accept" recommendation.

Reviewer #2: All comments have been addressed

2. Is the manuscript technically sound, and do the data support the conclusions?

Reviewer #2: Yes

3. Has the statistical analysis been performed appropriately and rigorously? 

Reviewer #2: Yes

4. Have the authors made all data underlying the findings in their manuscript fully available?

Reviewer #2: (No Response)

5. Is the manuscript presented in an intelligible fashion and written in standard English?

Reviewer #2: Yes

6. Review Comments to the Author

Reviewer #2: Thank you for addressing all requested revisions. The manuscript now includes a suitable synonym for "exploit," clarifies measures taken to reduce social desirability bias, cites specific AI guidelines from high-income countries, and provides examples of tools to detect AI content. Language improvements enhance clarity, and the discussion now explains age-related AI tool usage and international regulatory policies. These updates meet the review requirements, and I recommend the manuscript for acceptance.

7. PLOS authors have the option to publish the peer review history of their article (what does this mean?). If published, this will include your full peer review and any attached files.

Reviewer #2: **Yes: **Rajmohan Seetharaman

---

## [Editor Report · Acceptance letter]

5 Nov 2024

PONE-D-24-01790R2 

PLOS ONE

Dear Dr. Nantale, 

I'm pleased to inform you that your manuscript has been deemed suitable for publication in PLOS ONE. Congratulations! Your manuscript is now being handed over to our production team.

Kind regards, 

on behalf of

Dr. Timothy Omara 

Academic Editor

PLOS ONE